# Serum albumin is independently associated with higher mortality in adult sickle cell patients: Results of three independent cohorts

**Mehdi Nouraie**[1]*, **Allison E. Ashley-Koch**[2], **Melanie E. Garrett**[2], **Nithya Sritharan**[3], **Yingze Zhang**[1], **Jane Little**[4], **Victor R. Gordeuk**[5], **Mark T. Gladwin**[1,3], **Marilyn J. Telen**[2], **Gregory J. Kato**[1]

1 Department of Medicine, Vascular Medicine Institute, University of Pittsburgh, Pittsburgh, Pennsylvania, United States of America, 2 Department of Medicine, Duke University, Durham, North Carolina, United States of America, 3 Department of Medicine, University of Pittsburgh School of Medicine, Pittsburgh, Pennsylvania, United States of America, 4 Department of Medicine, University of North Carolina, Chapel Hill, North Carolina, United States of America, 5 Department of Medicine, University of Illinois at Chicago, Chicago, Illinois, United States of America

* nouraies@upmc.edu

**Data Availability Statement:** University of Pittsburgh and Duke University IRBs restrict data sharing for Walk-Phasst and SCD-OMG to

## Abstract

Sickle cell disease (SCD) impacts liver and kidney function as well as skin integrity. These complications, as well as the hyperinflammatory state of SCD, could affect serum albumin. Serum albumin has key roles in antioxidant, anti-inflammatory and antithrombotic pathways and maintains vascular integrity. In SCD, these pathways modulate disease severity and clinical outcomes. We used three independent SCD adult cohorts to assess clinical predictors of serum albumin as well its association with mortality. In 2553 SCD adult participants, the frequency of low (<35 g/L) serum albumin was 5%. Older age and lower hemoglobin (P <0.001) were associated with lower serum albumin in all three cohorts. In age and hemoglobin adjusted analysis, higher liver enzymes (P <0.05) were associated with lower serum albumin. In two of the three cohorts, lower kidney function as measured by Glomerular Filtration Rate (P<0.001) was associated with lower serum albumin. Lower serum albumin predicted higher risk of tricuspid regurgitation velocity $\geq$ 2.5 m/s (OR = 1.1 per g/L, P $\leq$0.01). In all three cohorts, patients with low serum albumin had higher mortality (adjusted HR $\geq$2.9, P $\leq$0.003). This study confirms the role of serum albumin as a biomarker of disease severity and prognosis in patients with SCD. Albumin as a biomarker and possible mediator of SCD severity should be studied further.

## Introduction

Albumin is multifunctional and one of the most abundant plasma proteins. In adults, the normal serum albumin level is 35–50 g/L, accounting for about 50% of the serum proteins [1]. Albumin is vital in human physiology because of its colloidal properties. It also plays a crucial

authorized investigators only. The data transfer agreement with NHLBI/BioLincc imposes restrictions on directly sharing data from CSSCD with any other investigator. Data from all three cohorts are regulated by strict data transfer agreement to protect the identity of patients with sickle cell disease. Walk-Phasst and SCD-OMG data are available at https://www.ncbi.nlm.nih.gov/gap/ for authorized investigators. CSSCD data are available at https://biolincc.nhlbi.nih.gov/home/. Walk-Phasst data can be requested from Ms. Nydia Chien (chienn@upmc.edu) and SCD-OMG can be requested from Ms. Radina Simeonova (radina.simeonova@duke.edu) for researchers who meet the criteria for access to confidential data. CSSCD data can be requested from Biolincc (biolincc@imsweb.com).

**Funding:** M.J. Telen and A.E. Ashley-Koch received grants from Doris Duke Charitable Foundation. The funder had no role in study design, data collection and analysis, decision to publish, or preparation of the manuscript.

**Competing interests:** The authors have declared that no competing interests exist.

role in maintaining osmotic pressure of extracellular fluids. Albumin has antioxidant properties and scavenges many plasma small molecules, including heme, and many drugs. These characteristics lead to its anti-inflammatory effect. In addition, albumin has also been identified to have emerging biologic roles in maintaining vascular integrity and even as an anti-thrombotic factor [2]. Albumin has a heparin-like activity and reduces platelet aggregation [3].

Serum albumin concentration is determined by its synthesis in the liver, its degradation, and its distribution between the intra- and extra-vascular spaces. The liver of a healthy human of 70 kg produces about 14 g of albumin a day. Albumin is eliminated from the kidney, gastro-intestinal tract and through catabolism, and has a total half-life of about 21 days [4] and intra-vascular half-life of less than 24 hours [1]. A decreased serum albumin level can be caused by low amino acid or energy supply, impaired liver function, increased albumin loss, catabolism or a change in distribution between intravascular and extravascular fluid [5]. Other predictors of decreased serum albumin are age, inflammation, chronic renal disease and general health status. In adults, serum albumin $<35$ g/L is considered low. Multiple studies suggest that there is a continuous association between lower serum albumin and higher mortality in patients with cancer [6], trauma and recent surgery [7], myocardial infarction (MI) [8], heart failure [9], stroke [10] and sepsis [11]. A recent analysis of healthy adults suggests that serum albumin is one of promising biomarker that can predict biological aging and can predict overall survival [12].

In patients with sickle cell disease (SCD), the presence of HbS in red blood cells initiates many harmful pathways that lead to vaso-occlusion, hemolytic anemia and down-stream reduced nitric oxide (NO) bioavailability, and increased inflammation, oxidative stress and platelet aggregation. In addition, SCD patients are at a higher risk of endothelial damage [13] and have a range of comorbidities, including liver and renal disease, that could affect their serum albumin [13]. Intravascular hemolysis releases free hemoglobin that undergoes oxidation to release free heme, which is partly carried by albumin. However, the role of serum albumin as a potential prognostic factor for mortality has not been examined in SCD. In this study of three independent adult SCD cohorts, we aim to assess the predictors of serum albumin and its prognostic value for mortality in adult SCD patients.

## Materials and methods

IRB of University of Pittsburgh and Duke approved the study.

### The treatment of pulmonary hypertension and sickle cell disease with sildenafil therapy (Walk-PHaSST)

The Walk-PHaSST study was designed to assemble a large cohort of patients with SCD and to screen their eligibility for a randomized clinical trial of sildenafil via echocardiographic and laboratory screening for pulmonary hypertension. In summary, the study recruited 720 adolescents and adults with SCD at 10 clinical centers in the United States (U.S.) and United Kingdom (UK) between 2007–2009. Each study participant was comprehensively evaluated by collecting clinical, laboratory and echocardiography data [14]. Serum albumin was measured at the time of recruitment for each study subject.

### Cooperative study of sickle cell disease (CSSCD)

The CSSCD was a prospective study of the clinical course of SCD in which more than 3700 children and adults were enrolled at 23 clinical centers throughout the continental U.S. between 1978–1988. All acute and chronic complications were documented at the

participating centers. Deaths were reported on a form that was completed by the center investigator [15]. Serum albumin was measured at the time of recruitment for each study subject.

### Outcome Modifying Genes in Sickle Cell Disease (OMG-SCD)

The OMG-SCD study enrolled over 800 adult patients with SCD to study genetic associations with clinical outcomes from five sickle cell centers in the southeastern U.S. between 2002 and 2015. All the participants' data included thorough medical histories, clinical assessments, laboratory testing of urine and blood samples, and genotyping [16]. Serum albumin was measured at the time of recruitment for each study subject.

### Statistical analysis

For this study, we selected adult ($\geq$18 years) patients with available serum albumin in each cohort. Linear regression was used to investigate the relationship between serum albumin and patient characteristics or clinical measures, adjusted for age and hemoglobin at enrollment. We used the hazard ratio from Cox regression analysis to assess the prognostic value of serum albumin at enrollment on mortality in each cohort, adjusting for age and hemoglobin at enrollment. In these models, the proportional hazard assumption was satisfied. Finally, logistic regression was used to test for association between serum albumin and elevated systolic pulmonary pressure as defined by tricuspid regurgitation velocity (TRV) $\geq$ 2.5 m/s), adjusting for age, lactate dehydrogenase (LDH), and serum creatinine. We adjusted for multiple hypotheses testing using Simes method [17].

## Results

### Predictors of serum albumin levels

**Walk-PHaSST.**   As an initial test cohort, this study provided data on 630 adults with SCD (93% Black). In these patients, lower hemoglobin level and older age were associated with lower serum albumin (S1 Table). In age- and hemoglobin-adjusted analysis, higher alkaline phosphatase, kidney dysfunction as measured by serum creatinine or estimated Glomerular Filtration Rate (eGFR) and markers of cardiopulmonary dysfunction, as measured by TRV and NT-proBNP, were associated with lower serum albumin (Table 1). Paradoxically, LDH was associated with higher serum albumin. Lower serum albumin was associated with greater risk of elevated systolic pulmonary artery pressure (adjusted OR = 1.10 per g/L, P < 0.001). Among these patients, 49 (7.8%) had serum albumin <35 g/L at steady state.

**CSSCD.**   As a validation cohort, data from 1303 adults with SCD (98% Black) from the CSSCD were analyzed. Replicating the findings from Walk-PHaSST, lower hemoglobin and older age were associated with lower serum albumin (S1 Table). In age- and hemoglobin-adjusted analysis, elevated transaminases and alkaline phosphatase were associated with lower serum albumin. Higher total bilirubin and LDH were also associated with higher serum albumin. Kidney function as defined by serum creatinine was not associated with serum albumin (Table 1). In this cohort, 36 (2.8%) had serum albumin <35 g/L.

**OMG-SCD.**   Providing a second validation cohort, the data from 620 adult SCD patients (99% Black) were evaluated from OMG-SCD. As observed in Walk-PHaSST and CSSCD, higher serum albumin was associated with higher hemoglobin and younger age. In this cohort, higher transaminases but not alkaline phosphatase were associated with lower serum albumin. Higher total bilirubin and better kidney function measured by eGFR were associated with higher albumin (Table 2). Lower serum albumin was associated with higher risk of elevated

**Table 1. Age and hemoglobin adjusted association between serum albumin and sickle cell clinical variables in adults with sickle cell disease in test cohorts.**

| | Walk-PHaSST | | CSSCD | |
| --- | --- | --- | --- | --- |
| | Adjusted beta (SE) or Mean (SE) | P value[1] | Adjusted beta (SE) or Mean (SE) | P value[1] |
| Female gender, n (%) | M = 41.8 (0.25); F = 41.4 (0.23) | 0.90 | M = 44.3 (0.18); F = 43.7 (0.16) | 0.15 |
| SS genotype, n (%) | SS/SB0 = 41.6 (0.20); Other = 41.5 (0.40) | 0.19 | SS/SB0 = 43.9 (0.17); Other = 43.9 (0.17) | 0.08 |
| Number of severe pains in last year, n (%) | 0.01 (0.36) | 0.36 | -- | -- |
| Chronic transfusion, n (%) | N = 41.5 (0.18); Y = 42.0 (0.49) | 0.91 | N = 43.7 (0.14); Y = 42.1 (1.23) | 0.19 |
| History of acute chest syndrome, n (%) | N = 41.6 (0.28); Y = 41.6 (0.21) | 0.59 | -- | -- |
| Leg ulcer, n (%) | N = 41.9 (0.18); Y = 41.3 (0.38) | 0.40 | N = 44.1 (0.17); Y = 43.8 (0.17) | 0.34 |
| BMI ($kg/m^2$) | -0.03 (0.02) | 0.14 | -0.01 (0.01) | 0.15 |
| MCV (fL) | -0.01 (0.01) | 0.42 | -0.008 (0.01) | 0.49 |
| White blood cell count ($x10^9$/L) | 0.04 (0.047) | 0.37 | -0.07 (0.04) | 0.045 |
| Platelet count ($x10^9$/L) | -0.0002 (0.001) | 0.88 | -0.0004 (0.001) | 0.66 |
| Lactate dehydrogenase (U/L) | 0.003 (0.001) | <0.001 | 0.002 (0.001) | <0.001 |
| Reticulocyte count ($x10^9$/L) | 0.03 (0.01) | 0.020 | -0.02 (0.01) | 0.015 |
| Total bilirubin (mg/dL) | -0.01 (0.06) | 0.87 | 0.18 (0.05) | 0.001 |
| Alanine aminotransferase (U/L) | -0.01 (0.008) | 0.20 | -0.01 (0.003) | 0.002 |
| Aspartate aminotransferase (U/L) | -0.003 (0.006) | 0.55 | -0.008 (0.002) | 0.002 |
| Alkaline phosphatase (U/L) | -0.01 (0.003) | <0.001 | -0.01 (0.002) | <0.001 |
| Creatinine (mg/dL) | -0.77 (0.19) | <0.001 | 0.06 (0.18) | 0.76 |
| eGFR ($mL/min/1.73m^2$) | 0.02 (0.006) | <0.001 | 0.007 (0.004) | 0.088 |
| NT-proBNP (pg/mL) | -0.0003 (0.0001) | <0.001 | -0.00004 (0.00004) | 0.38 |
| Tricuspid regurgitation velocity, m/sec | -2.2 (0.45) | <0.001 | -- | -- |

[1] P values $\leq$ 0.015 are significant after correcting for multiple hypothesis testing using False Discover Rate <0.05.

systolic pulmonary artery pressure (adjusted OR = 1.09 per g/L, P = 0.0119). In this cohort, 49 (7.9%) had serum albumin <35 g/L.

In both Walk-PHaSST and OMG cohorts, higher eGFR was consistently associated with higher serum albumin whereas in CSSCD and Walk-PHaSST cohorts, higher LDH and lower alkaline phosphatase were associated with higher serum albumin.

## Serum albumin is associated with higher mortality

**Walk-PHaSST.** We followed 592 patients for a median of 29 months (IQR: 25–33). There were 21 (3.6%) patients who died over the course of this follow-up period. Serum albumin as a continuous variable (HR = 0.91, 95% CI: 0.85–0.97, P = 0.004) was inversely associated with mortality and low serum albumin (HR = 5.3; 95% CI: 2.0–13.6, P = 0.001) was significantly associated with higher mortality. Probability of survival at 1 and 3 years was 99% and 96% respectively in patients with normal albumin compared to 89% and 86% respectively in patients with low albumin. After adjusting for age and hemoglobin, continuous serum albumin (HR = 0.91, 95% CI: 0.85–0.97, P = 0.014) and low serum albumin (HR = 4.7, 95% CI: 1.7–13.0, P = 0.003) were both significantly associated with mortality (Fig 1A).

**CSSCD.** One thousand thirty-one patients were followed for a median of 82 months (IQR: 60–89) and 237 (23.0%) patients died during the follow-up period. Continuous serum albumin (HR = 0.94, 95% CI: 0.92–0.96, P < 0.001) and low serum albumin (HR = 3.5, 95% CI: 2.1–5.9, P < 0.001) were both significantly associated with mortality. Probability of survival at 1 and 5 years was 97% and 89% in patients with normal albumin compared to 93% and 67% in patients with low albumin. After adjusting for age and hemoglobin, both continuous serum

**Table 2. Age and hemoglobin adjusted association between serum albumin and sickle cell clinical variables in adults with sickle cell disease in validation cohort (OMG).**

| | Adjusted mean (SE) or beta (SE) | P value[1] |
|---|---|---|
| Female gender | F = 41.3 (0.27), M = 41.6 (0.29) | 0.39 |
| SS genotype | SS/SB0 = 41.4 (0.21), other = 41.4 (0.66) | 0.99 |
| Hospitalizations for severe pain in last year | 0–1 = 41.7 (0.27), | 0.036 |
| | 2–4 = 40.5 (0.40), | |
| | >4 = 41.7 (0.50) | |
| Chronic transfusion | N = 41.3 (0.21), Y = 41.3 (0.80) | 0.97 |
| History of acute chest syndrome | N = 41.6 (0.39), Y = 41.2 (0.24) | 0.41 |
| Leg ulcer | N = 41.5 (0.24), Y = 41.0 (0.44) | 0.33 |
| BMI (kg/m$^2$) | -0.064 (0.037) | 0.08 |
| Current MCV (fL) | -0.019 (0.014) | 0.18 |
| Current WBC (x10$^9$/L) | 0.10 (0.05) | 0.040 |
| Current Platelet (x10$^9$/L) | -0.002 (0.001) | 0.10 |
| Lactate dehydrogenase (U/L) | -0.0007 (0.001) | 0.49 |
| Reticulocyte count (x10$^9$/L) | 0.001 (0.002) | 0.53 |
| Total bilirubin (mg/dL) | 0.20 (0.08) | 0.009 |
| Alanine aminotransferase (U/L) | -0.015 (0.007) | 0.049 |
| Aspartate aminotransferase (U/L) | -0.013 (0.006) | 0.022 |
| Alkaline phosphatase (U/L) | -0.006 (0.003) | 0.08 |
| Creatinine (mg/dL) | -0.37 (0.20) | 0.07 |
| eGFR (mL/min/1.73m$^2$) | 0.021 (0.006) | <0.001 |
| NT-proBNP (pg/mL) | -0.0003 (0.0011) | 0.78 |
| Tricuspid regurgitation velocity (m/sec) | -0.65 (0.39) | 0.10 |

[1] P values ≤ 0.001 are significant after correcting for multiple hypothesis testing using False Discover Rate <0.05.

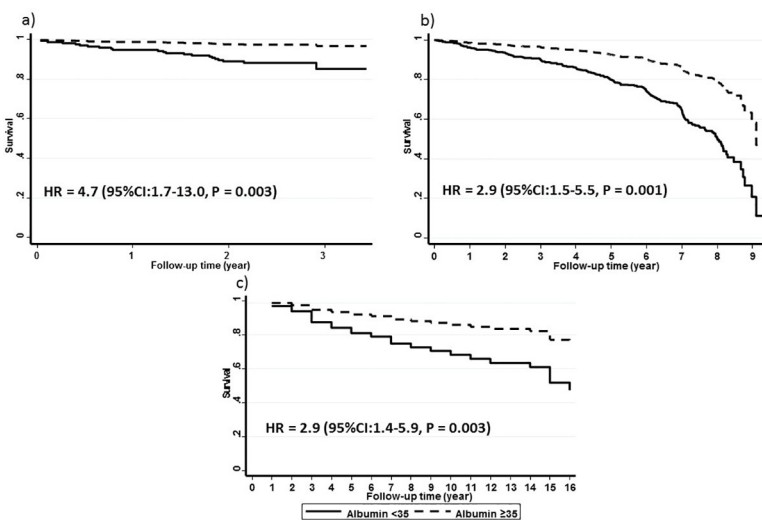

**Fig 1.** Low serum albumin (<35 g/L solid line) predicts higher mortality among sickle cell disease patients in a) Walk-PHaSST b) CSSCD and c) OMG-SCD cohorts. Graphs show the age and hemoglobin adjusted survival in each cohort.

albumin (HR = 0.97, 95% CI: 0.93–0.998, P = 0.039) and low serum albumin (HR = 2.9, 95% CI: 1.5–5.5, P = 0.001) were significantly associated with mortality (Fig 1B).

**OMG-SCD.** We followed 195 SCD patients for a median of 156 months (IQR: 108–192). In that time, 53 patients died (27.2%). Serum albumin both as a continuous variable (HR = 0.88, 95% CI:0.84–0.93, P < 0.001) and as a dichotomous variable (<35 g/L; HR = 3.8, 95% CI: 2.0–7.2, P < 0.001) was significantly associated with mortality. Probability of survival at 1 and 5 years was 98% and 90% in patients with normal albumin compared to 79% and 63% in patients with low albumin. After adjusting for age and hemoglobin, both serum albumin as a continuous variable (HR = 0.92, 95% CI: 0.87–0.98, P = 0.008) and lower serum albumin (HR = 2.9, 95% CI: 1.4–5.9, P = 0.003) were significantly associated with mortality (Fig 1c).

## Discussion

In our study of three independent cohorts of adults with SCD, serum albumin concentration was significantly associated with severity of anemia, as well as liver and renal dysfunction. Low serum albumin was observed in ~5% of patients and predicted a higher estimated systolic pulmonary artery pressure and overall mortality of patients.

Sickle cell disease is a hyper-inflammatory state in which inflammatory markers and reactive oxygen species (ROS) are increased. Serum concentration of pro-inflammatory cytokines, such as IL-6, that are associated with decreased albumin production, is higher in SCD patients [18]. Excessive production of ROS and free heme in SCD may also contribute to a higher rate of albumin catabolism [13].

Acute and chronic complications of SCD impact hepatobiliary function. Sickle cell hepatopathy includes a range of manifestations ranging from liver dysfunction to cholestasis [19]. Liver dysfunction could impact albumin production in SCD patients. In this study, we have been able to replicate an association between decreased hepatobiliary function (as measured by liver transaminases or ALK) and lower serum albumin in three independent SCD cohorts. Liver function can also influence other heme carrying proteins including hemopexin and haptoglobin. We did not measure these proteins in all cohorts but an analysis in the Walk-PHaSST cohort showed that serum albumin and haptoglobin are not correlated (r = 0.06, P = 0.17). Studies in children with severe malnutrition have shown that plasma albumin concentration was modulated by catabolic rates rather than synthesis rates [20, 21]. SCD patient are in a state of hyper-inflammation which could lead to faster catabolic rates.

Renal dysfunction, as measured by higher serum creatinine and lower eGFR, was a strong predictor of low serum albumin in walk-PHaSST and OMG cohorts. Renal insufficiency is a frequent comorbidity in SCD patients, with a prevalence of >25% in adult patients [22]. It is also an important prognostic factor in these patients [23]. Renal tubular loss of albumin and high urine albumin to creatinine ratio is observed in over one third of adults with SCD [24]. Low albumin predicts mortality in patients with chronic kidney disease [25, 26]. Serum albumin is considered a marker of general nutrition and health status in patients with chronic kidney disease. In addition, this prognostic effect could also be explained by the antioxidant effect of albumin and its protection against metabolic acidosis and further kidney damage [27, 28].

In our study, both elevated TRV and NT-proBNP, which are non-invasive markers of elevated pulmonary systolic blood pressure and left ventricular (LV) diastolic dysfunction, are two major prognostic factors in adults with SCD [23]. Both were significantly correlated with low serum albumin. Lower serum albumin has also been associated with worse LV diastolic dysfunction in children with chronic kidney disease [29]. Serum albumin level is an independent predictor of one-year survival in patients with heart failure with preserved ejection fraction [30]. Low-grade albuminuria has been associated with lower LV diastolic function in

other clinical settings [31]. The Heart Outcomes Prevention Evaluation study suggests that higher urine creatinine to albumin ratio within the normal range is significantly associated with the increasing prevalence of myocardial dysfunction and mortality [32]. The effect of serum albumin on myocardial function is potentially mediated by multiple mechanisms. Albumin maintains the integrity of the microvasculature of the myocardium and is protective against myocardial edema. Also, it protects against myocyte oxidative stress and inflammation [4, 28, 33]. In addition to such direction protection, this association may be explained by the role of chronic liver disease in the development pulmonary hypertension through portal hypertension [34].

It is also conceivable that low albumin could mediate SCD pathophysiologic mechanisms. It is well known that free heme released during hemolysis is a part of SCD pathophysiology, with increasing evidence that heme induces inflammatory pathways [35]. Heme is hydrophobic and insoluble in plasma and thus is carried by plasma proteins, including haptoglobin, hemopexin and albumin. Serum albumin has a lower affinity for free heme than haptoglobin or hemopexin but can sequester the heme when these two proteins are depleted or saturated [36]. One provocative hypothesis is that low serum albumin provides low heme carrying capacity, potentially increasing the concentrations of free heme that might be taken up by susceptible tissues. Heme response in neutrophils, macrophages, erythroid cells and endothelial cells plays a role in SCD pathophysiology [37].

There are some limitations in our study. We have not measured the nutrition status of our patients, which could provide some insight regarding the cause of the low albumin. In addition, we have not identified the specific cause of mortality. Thus, we cannot speculate whether the low albumin led to specific causes of death or represents a general risk factor for mortality. We also did not examine proteinuria in these cohorts. Serum albumin was comparable between all three cohorts whereas inter-study variation exists in some other clinical variables. This variability could be due to random variation or different patient characteristics including age, disease severity, genetic structure, disease modifying treatments (including hydroxyurea and chronic transfusion) and comorbidities (as well as their treatment). Other factors that can cause inter-study variability are different birth cohorts and time period between CSSCD and the two other studies, differences in the study protocol and laboratory measurements. These differences could limit the generalizability of our findings to any new sickle cell cohort. However, the inclusion of two replication cohorts provides more confidence in our findings. To the best of our knowledge, this is the first report indicating predictors and prognostic impact of serum albumin in SCD patients, despite the ready availability of serum albumin measurement in clinical settings. Understanding the link between inflammation, hypoalbuminemia, and poor outcome in SCD patients could help identify at-risk patients and define novel mechanistic applications of serum albumin in SCD.

There are existing controversies to consider albumin as a supplementary treatment for liver and renal failure as well as volume resuscitation in sepsis [38]. Our findings support the importance of plasma albumin measure and potential benefit of boosting plasma levels of albumin in SCD, especially in SCD endemic countries with high frequency of protein-calorie malnutrition.

## Supporting information

**S1 Table. Unadjusted correlation between serum albumin and clinical variables in adults with sickle cell disease in test cohorts.** Results are in median (IQR) unless otherwise specified.
(DOCX)

**S2 Table. Unadjusted correlation between serum albumin and sickle cell clinical variables in adults with sickle cell disease in validation cohort (OMG).** Results are in median (IQR) unless otherwise specified.
(DOCX)

## Author Contributions

**Conceptualization:** Mehdi Nouraie, Allison E. Ashley-Koch, Victor R. Gordeuk, Mark T. Gladwin, Marilyn J. Telen, Gregory J. Kato.

**Data curation:** Allison E. Ashley-Koch, Nithya Sritharan, Yingze Zhang, Jane Little, Victor R. Gordeuk, Mark T. Gladwin, Marilyn J. Telen, Gregory J. Kato.

**Formal analysis:** Mehdi Nouraie, Melanie E. Garrett, Gregory J. Kato.

**Funding acquisition:** Allison E. Ashley-Koch, Mark T. Gladwin, Marilyn J. Telen.

**Investigation:** Mehdi Nouraie, Allison E. Ashley-Koch, Marilyn J. Telen.

**Methodology:** Mehdi Nouraie, Allison E. Ashley-Koch, Melanie E. Garrett, Marilyn J. Telen, Gregory J. Kato.

**Writing – original draft:** Mehdi Nouraie, Allison E. Ashley-Koch, Marilyn J. Telen, Gregory J. Kato.

**Writing – review & editing:** Mehdi Nouraie, Allison E. Ashley-Koch, Victor R. Gordeuk, Mark T. Gladwin, Marilyn J. Telen, Gregory J. Kato.

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
