## [Decision Letter · Decision Letter 0]

27 May 2020

PONE-D-20-14109

Serum albumin independently predicts higher mortality in adult sickle cell patients: Results of three independent cohorts.

PLOS ONE

Dear Dr. Mehdi Nouraie,

Thank you for submitting your manuscript to PLOS ONE. After careful consideration, we feel that it has merit but does not fully meet PLOS ONE’s publication criteria as it currently stands. Therefore, we invite you to submit a revised version of the manuscript that addresses the points raised during the review process.

The conclusion could have been bolstered by more references to the literature on albumin synthesis rates in  children with acute-phase protein responses to infection in edematous and non-edematous protein-energy malnutrition. The speculation that hepatic synthetic rates may be affected because of the the association of low albumin with liver enzyme elevation raised concern in the concluding paragraph which should be re-phrased.

We look forward to receiving your revised manuscript.

Kind regards,

Mary Hamer Hodges

Academic Editor

PLOS ONE

Journal Requirements:

Additional Editor Comments (if provided):

We recommend addressing the comments made by Reviewer #1 and resubmitting.

Reviewers' comments:

Reviewer's Responses to Questions

**Comments to the Author**

1. Is the manuscript technically sound, and do the data support the conclusions?

Reviewer #1: Yes

2. Has the statistical analysis been performed appropriately and rigorously? 

Reviewer #1: Yes

3. Have the authors made all data underlying the findings in their manuscript fully available?

Reviewer #1: Yes

4. Is the manuscript presented in an intelligible fashion and written in standard English?

Reviewer #1: Yes

5. Review Comments to the Author

Reviewer #1: This paper reports on the outcome of analysis on 3 large retrospective cohort of subjects with Sickle Cell Disease. The hypotheses, and justification were clear. The results were clear. Albumin as a negative acute phase protein is a biomarker for worse outcome and this is consistent across a range of clinical models. The conclusion could have been bolstered by reference to the literature where albumin synthesis rates were measured in vivo. So in a model of severe malnutrition in children albumin concentration were modulated by catabolic rates rather than synthesis rates. In fact children with severe malnutrition were able to synthesize a range of positive acute phase proteins at fast enough rate in response to infections. So Sickle cell as an example of inflammatory state may behave similarly. (Morlese JF, Forrester T, Badaloo A, Del Rosario M, Frazer M, Jahoor F. Albumin kinetics in edematous and nonedematous protein-energy malnourished children. Am J Clin Nutr. 1996 Dec;64(6):952-9. doi: 10.1093/ajcn/64.6.952. PMID: 8942422 & Reid, M., Badaloo, A., Forrester, T., Morlese, J.F., Heird, W.C. & Jahoor, F. (2002) The acute-phase protein response to infection in edematous and nonedematous protein-energy malnutrition. Am J Clin Nutr, 76, 1409-1415.). While the authors speculated in the manuscript that hepatic synthetic rates may be affected because of the the association of low albumin with liver enzyme elevation this may not be the case. Within that context I am little concerned about the wording of the concluding paragraph which at a casual glance may seem to be advocating supplemental albumin treatment in SCD and perhaps a clear definite statement that this is not their recommendation should be made

6. PLOS authors have the option to publish the peer review history of their article (what does this mean?). If published, this will include your full peer review and any attached files.

Reviewer #1: Yes: Marvin E Reid

---

## [Author Response · Author response to Decision Letter 0]

4 Jun 2020

PONE-D-20-14109

Dear Dr. Hodges;

We appreciate your and review comments on our manuscript. Hereby, we are responding to comments and addressing changes in the manuscript: 

Comment: The conclusion could have been bolstered by more references to the literature on albumin synthesis rates in children with acute-phase protein responses to infection in edematous and non-edematous protein-energy malnutrition. The speculation that hepatic synthetic rates may be affected because of the association of low albumin with liver enzyme elevation raised concern in the concluding paragraph which should be re-phrased.

Response: We modified the discussion and conclusion to reflect the reviewer’s comments. 

Comment: In your Data Availability statement, you have not specified where the minimal data set underlying the results described in your manuscript can be found. PLOS defines a study's minimal data set as the underlying data used to reach the conclusions drawn in the manuscript and any additional data required to replicate the reported study findings in their entirety. All PLOS journals require that the minimal data set be made fully available. For more information about our data policy, please see http://journals.plos.org/plosone/s/data-availability.

Response: The current manuscript is using three different databases. Data from one of these cohort (CSSCD) was received through an agreement with NHLBI/BioLincc. This agreement prohibits investigator form any further data sharing. However, other investigators can apply directly to BioLincc for their own access to the data. The remaining two cohorts are governed by academic institutional IRBs which have their own restrictions on sharing. However, both of the other cohorts are available for access via dbGaP. So, regretfully we are unable to comply within the legal terms of our data use agreement.

Comment: We note that you have included the phrase “data not shown” in your manuscript. Unfortunately, this does not meet our data sharing requirements. PLOS does not permit references to inaccessible data. We require that authors provide all relevant data within the paper, Supporting Information files, or in an acceptable, public repository. Please add a citation to support this phrase or upload the data that corresponds with these findings to a stable repository (such as Figshare or Dryad) and provide and URLs, DOIs, or accession numbers that may be used to access these data. Or, if the data are not a core part of the research being presented in your study, we ask that you remove the phrase that refers to these data.

Response: This sentence was changed to: 

 “We did not measure these proteins in all cohorts but an analysis in the Walk-PHaSST cohort showed that serum albumin and haptoglobin are not correlated (r = 0.06, P = 0.17).”

Reviewer #1 comment: This paper reports on the outcome of analysis on 3 large retrospective cohort of subjects with Sickle Cell Disease. The hypotheses, and justification were clear. The results were clear. Albumin as a negative acute phase protein is a biomarker for worse outcome and this is consistent across a range of clinical models. The conclusion could have been bolstered by reference to the literature where albumin synthesis rates were measured in vivo. So in a model of severe malnutrition in children albumin concentration were modulated by catabolic rates rather than synthesis rates. In fact children with severe malnutrition were able to synthesize a range of positive acute phase proteins at fast enough rate in response to infections. So Sickle cell as an example of inflammatory state may behave similarly. (Morlese JF, Forrester T, Badaloo A, Del Rosario M, Frazer M, Jahoor F. Albumin kinetics in edematous and nonedematous protein-energy malnourished children. Am J Clin Nutr. 1996 Dec;64(6):952-9. doi: 10.1093/ajcn/64.6.952. PMID: 8942422 & Reid, M., Badaloo, A., Forrester, T., Morlese, J.F., Heird, W.C. & Jahoor, F. (2002) The acute-phase protein response to infection in edematous and nonedematous protein-energy malnutrition. Am J Clin Nutr, 76, 1409-1415.). While the authors speculated in the manuscript that hepatic synthetic rates may be affected because of the association of low albumin with liver enzyme elevation this may not be the case. Within that context I am little concerned about the wording of the concluding paragraph which at a casual glance may seem to be advocating supplemental albumin treatment in SCD and perhaps a clear definite statement that this is not their recommendation should be made

Response: We cited these references in discussion as:

 “Studies in children with severe malnutrition have shown that plasma albumin concentration was modulated by catabolic rates rather than synthesis rates[19, 20]. SCD patient are in a state of hyper-inflammation which could lead to faster catabolic rates.”

Also, we changed the conclusion to:

 “There are existing controversies to consider albumin as a supplementary treatment for liver and renal failure as well as volume resuscitation in sepsis [37]. Our findings support the importance of plasma albumin measure and potential benefit of boosting plasma levels of albumin in SCD, especially in SCD endemic countries with high frequency of protein-calorie malnutrition.”

 Sincerely yours,

Mehdi Nouraie, MD, PhD

---

## [Decision Letter · Decision Letter 1]

9 Jul 2020

PONE-D-20-14109R1

Serum albumin independently predicts higher mortality in adult sickle cell patients: Results of three independent cohorts .

PLOS ONE

Dear Mehdi Nouraie,

Thank you for submitting your manuscript to PLOS ONE. After careful consideration, we feel that it has merit but does not fully meet PLOS ONE’s publication criteria as it currently stands. Therefore, we invite you to submit a revised version of the manuscript that addresses the points raised during the review process.

Please address the inter-study variability, possible causes and study limitations

We look forward to receiving your revised manuscript.

Kind regards,

Mary Hamer Hodges

Academic Editor

PLOS ONE

Additional Editor Comments (if provided):

Please address the comments from reviewer #3 regarding the inter-study variability. If no potential cause for this variability can be identified it should be discussed as a weakness of their results.

Reviewers' comments:

Reviewer's Responses to Questions

**Comments to the Author**

1. If the authors have adequately addressed your comments raised in a previous round of review and you feel that this manuscript is now acceptable for publication, you may indicate that here to bypass the “Comments to the Author” section, enter your conflict of interest statement in the “Confidential to Editor” section, and submit your "Accept" recommendation.

Reviewer #1: All comments have been addressed

Reviewer #2: All comments have been addressed

Reviewer #3: (No Response)

2. Is the manuscript technically sound, and do the data support the conclusions?

Reviewer #1: Yes

Reviewer #2: Yes

Reviewer #3: Yes

3. Has the statistical analysis been performed appropriately and rigorously? 

Reviewer #1: Yes

Reviewer #2: Yes

Reviewer #3: Yes

4. Have the authors made all data underlying the findings in their manuscript fully available?

Reviewer #1: Yes

Reviewer #2: Yes

Reviewer #3: Yes

5. Is the manuscript presented in an intelligible fashion and written in standard English?

Reviewer #1: Yes

Reviewer #2: Yes

Reviewer #3: Yes

6. Review Comments to the Author

Reviewer #1: The authors have addressed my concerns appropriately. The limitations of post-hoc secondary data analyses were addressed satisfactorily.

Reviewer #2: Serum albumin independently predicts higher mortality in adult sickle cell patients:

Results of three independent cohorts is an interesting article all modifications were done as per reviewers recommendations

the manuscript looks much better

Reviewer #3: This paper presents the results for 3 different SSD studies. It was found that low serum albumin was associated with higher mortality across all 3 studies. However, the paper also looks at predictors of low serum albumin in Tables 1 and 2 of the manuscript. There is a large amount of inter-study variability which is not well accounted for. It seems possible that there are significant differences in the enrolled populations of the 3 studies and that these differences may impact the predictors relationship with serum albumin. Please see the attached PDF for my specific comments.

7. PLOS authors have the option to publish the peer review history of their article (what does this mean?). If published, this will include your full peer review and any attached files.

Reviewer #1: **Yes: **MARVIN REID

Reviewer #2: **Yes: **Mohamed A Yassin

Reviewer #3: No

---

## [Author Response · Author response to Decision Letter 1]

12 Jul 2020

Dear Dr. Hodges

Authors appreciate your and reviewers’ invaluable new comments on our manuscript. Hereby, we are addressing the comments and changes in the new version. 

Editor Review

Please address the inter-study variability, possible causes and study limitations

Response: We added this statement to study limitation:

“Serum albumin was comparable between all three cohorts whereas inter-study variation exists in some other clinical variables. This variability could be due to random variation or different patient characteristics including age, disease severity, genetic structure, disease modifying treatments (including hydroxyurea and chronic transfusion) and comorbidities (as well as their treatment). Other factors that can cause inter-study variability are different birth cohorts and time period between CSSCD and the two other studies, differences in the study protocol and laboratory measurements. These differences could limit the generalizability of our findings to any new sickle cell cohort.”

Reviewer #3 comments: 

This paper presents the results for 3 different SSD studies. It was found that low serum albumin was associated with higher mortality across all 3 studies. However, the paper also looks at predictors of low serum albumin in Tables 1 and 2 of the manuscript. There is a large amount of inter-study variability which is not well accounted for. It seems possible that there are significant differences in the enrolled populations of the 3 studies and that these differences may impact the predictors relationship with serum albumin. 

See below for specific comments.

1.1 Specific Comments

1. The term “predicts” in the title is misleading. Many people will interpret the claim that “Serum albumin independently predicts higher mortality” as a statement of that serum albumin has a high predictive value (sensitivity, specificity, AUC, etc) with respect to mortality. However, what is shown is that low serum corresponds to a high hazard ratio. I think “Serum albumin is independently associated with higher mortality” is a more appropriate phrase.

Response: We changed the title to: 

 “Serum albumin is independently associated with higher mortality in adult sickle cell patients: Results of three independent cohorts”

2. You mentioned you tested serum albumin for normality. Do you mean the residuals from the individual fitted linear models are normal? Serum albumin should not be normal on its own, unless the predictor variables are all insignificant. The residuals are what we expect to be normal. In point of fact, when the sample size is large the ordinary least squares estimates are asymptotically normal even if the residuals are non-normal. The sample sizes in this study are large enough that I think the results are valid even if the residuals are not normal.

Response: We removed this statement from the statistical analysis as it did not impact our analysis. 

3. The false discovery rate adjustment method and procedure should be stated.

Response: 

We added this statement in statistical method:

“We adjusted for multiple hypotheses testing using Simes method [17]”.

4. The results of table 1 and 2 make drawing conclusions about associations between serum albumin and sickle cell clinical variables challenging. Many of the variables change signs, even significantly, from study to study. For example ARC beta is significant in the WALK study at 0.03. For CCSSD it is significant at -0.02. Then it is insignificant at 0.001 for OMG. There are many examples of this behavior. Inter-study variability of beta and even significance is to expected; however, when a variable is significant and changes signs from study to study

represents a failure in validation. A disclaimer or a summary across all 3 studies of which variables of interest exhibited consistent behavior would help a reader understand how to better interpret Tables 1 and 2.

Response: We added this statement at the end of first result section:

“In both Walk-PHaSST and OMG cohorts, higher eGFR was consistently associated with higher serum albumin whereas in CSSCD and Walk-PHaSST cohorts, higher LDH and lower alkaline phosphatase were associated with higher serum albumin.” 

5. Is there any insight into why there is such a significant amount of inter-study variability. Certainly CCSSCD, which ended in 1988, might be difficult to compare to patients collected after the year 2000. Are there differences in Age, hemoglobin, or other variables appearing in tables 1 and 2 between the 3 studies? Perhaps the populations at enrollment are significantly different. If so multivariate models that include the variables in table 1 and 2 might be more consistent.

Response: We added some statements to describe inter-study variability (please see the response to editor comment). In addition, tables 1 and 2 are presenting age and hemoglobin adjusted correlations to partially adjust for baseline differences between three cohorts. 

6. In Tables 1 and 2 it is stated that p<0.015 and <0.001 are significant after FDR. However, there are rows in bold with p-values greater than these amounts.

Response: We removed the bold marks to prevent any confusion in these tables. 

7. In the discussion it is stated that “ Finally, we also did 203 not correct for multiple testing.” However, in each table the FDR rate is listed. FDR is a multiplicity adjustment method. This is confusing was and FDR used or not? 

Response: It was removed from discussion 

Sincerely,

Mehdi Nouraie, MD, PhD

Associate Professor of Medicine

University of Pittsburgh

---

## [Decision Letter · Decision Letter 2]

29 Jul 2020

Serum albumin is independently associated with higher mortality in adult sickle cell patients: Results of three independent cohorts

PONE-D-20-14109R2

Dear Dr. Mehdi Nouraie,

We’re pleased to inform you that your manuscript has been judged scientifically suitable for publication and will be formally accepted for publication once it meets all outstanding technical requirements.

Kind regards,

Mary Hamer Hodges

Academic Editor

PLOS ONE

Additional Editor Comments (optional):

Reviewers' comments:

Reviewer's Responses to Questions

**Comments to the Author**

1. If the authors have adequately addressed your comments raised in a previous round of review and you feel that this manuscript is now acceptable for publication, you may indicate that here to bypass the “Comments to the Author” section, enter your conflict of interest statement in the “Confidential to Editor” section, and submit your "Accept" recommendation.

Reviewer #3: All comments have been addressed

2. Is the manuscript technically sound, and do the data support the conclusions?

Reviewer #3: Yes

3. Has the statistical analysis been performed appropriately and rigorously? 

Reviewer #3: Yes

4. Have the authors made all data underlying the findings in their manuscript fully available?

Reviewer #3: Yes

5. Is the manuscript presented in an intelligible fashion and written in standard English?

Reviewer #3: Yes

6. Review Comments to the Author

Reviewer #3: (No Response)

7. PLOS authors have the option to publish the peer review history of their article (what does this mean?). If published, this will include your full peer review and any attached files.

Reviewer #3: **Yes: **Adam Lane

---

## [Editor Report · Acceptance letter]

30 Jul 2020

PONE-D-20-14109R2 

Serum albumin is independently associated with higher mortality in adult sickle cell patients: Results of three independent cohorts 

Dear Dr. Nouraie:

I'm pleased to inform you that your manuscript has been deemed suitable for publication in PLOS ONE. Congratulations! Your manuscript is now with our production department. 

Kind regards, 

on behalf of

Dr. Mary Hamer Hodges 

Academic Editor

PLOS ONE